# Heparin and Gelatin Co-Functionalized Polyurethane Artificial Blood Vessel for Improving Anticoagulation and Biocompatibility

**DOI:** 10.3390/bioengineering12030304

**Published:** 2025-03-18

**Authors:** Jimin Zhang, Jingzhe Guo, Junxian Zhang, Danting Li, Meihui Zhong, Yuxuan Gu, Xiaozhe Yan, Pingsheng Huang

**Affiliations:** 1Hebei Key Laboratory of Functional Polymers, School of Chemical Engineering and Technology, Hebei University of Technology, Tianjin 300130, China; zhangjimin@hebut.edu.cn (J.Z.); 18251836002@163.com (J.G.); 202221501028@stu.hebut.edu.cn (J.Z.); 18830490298@163.com (D.L.); 15864081918@163.com (M.Z.); guyuxuan0a@163.com (Y.G.); 13483017919@163.com (X.Y.); 2Tianjin Key Laboratory of Biomaterial Research, Institute of Biomedical Engineering, Chinese Academy of Medical Sciences and Peking Union Medical College, Tianjin 300192, China

**Keywords:** polyurethane, multiple hydrogen bonds, low-molecular-weight heparin, gelatin, artificial blood vessel

## Abstract

The primary challenges in the tissue engineering of small-diameter artificial blood vessels include inadequate mechanical properties and insufficient anticoagulation capabilities. To address these challenges, urea-pyrimidone (Upy)-based polyurethane elastomers (PIIU-B) were synthesized by incorporating quadruple hydrogen bonding within the polymer backbone. The synthesis process employed poly(L-lactide-ε-caprolactone) (PLCL) as the soft segment, while di-(isophorone diisocyanate)-Ureido pyrimidinone (IUI) and isophorone diisocyanate (IPDI) were utilized as the hard segment. The resulting PIIU-B small-diameter artificial blood vessel with a diameter of 4 mm was fabricated using the electrospinning technique, achieving an optimized IUI/IPDI composition ratio of 1:1. Enhanced by multiple hydrogen bonds, the vessels exhibited a robust elastic modulus of 12.45 MPa, an extracellular matrix (ECM)-mimetic nanofiber morphology, and a high porosity of 41.31%. Subsequently, the PIIU-B vessel underwent dual-functionalization with low-molecular-weight heparin and gelatin via ultraviolet (UV) crosslinking (designated as PIIU-B@LHep/Gel), which conferred superior biocompatibility and exceptional anticoagulation properties. The study revealed improved anti-platelet adhesion characteristics as well as a prolonged activated partial thromboplastin time (APTT) of 157.2 s and thrombin time (TT) of 64.2 s in vitro. Following a seven-day subcutaneous implantation, the PIIU-B@LHep/Gel vessel exhibited excellent biocompatibility, evidenced by complete integration with the surrounding peri-implant tissue, significant cell infiltration, and collagen formation in vivo. Consequently, polyurethane-based artificial blood vessels, reinforced by multiple hydrogen bonds and dual-functionalized with heparin and gelatin, present as promising candidates for vascular tissue engineering.

## 1. Introduction

Cardiovascular disease (CVD) is predominantly attributed to atherosclerosis, a leading cause of mortality globally, accounting for approximately 523 million deaths [1,2]. The most prevalent clinical intervention is bypass surgery, wherein a vascular graft is employed to restore normal blood flow [3,4]. Typically, arteries from the chest wall and veins from the legs, which do not elicit immune rejection, are preferred [5]. However, the availability of these sources is severely constrained [6]. Artificial blood vessels, derived from abundant resources, present viable alternatives [7,8,9,10]. For instance, artificial blood vessels made from expanded polytetrafluoroethylene (ePTFE) and polyethylene terephthalate (PET) have been successfully utilized in millions of patients [11,12,13]. Nevertheless, their chemical inertness and limited compliance restrict their use in large-diameter vessels (diameter > 6 mm) [14,15], while their ineffectiveness in small-diameter vessels (diameter < 6 mm) is attributed to challenges in degradation and a propensity for thrombosis [16,17].

Synthetic polymers, such as polyesters, polyethers, and polyurethane (PU), have been extensively utilized in the fabrication of vascular stents due to their advantageous properties, including excellent biocompatibility, suitable degradation rates, robust thermal stability, and well-established manufacturing techniques [17,18]. Among these materials, polyurethane-based artificial blood vessels are notable for their high mechanical strength and elasticity, which can be tailored by modifying the ratio of soft segments to hard segments [19]. Polymers based on polyesters and polyethers, such as polycaprolactone (PCL), poly(L-lactide-co-ε-caprolactone) (PLCL), polyglycolic acid (PGA), and polyethylene glycol (PEG), are valued for their desirable elastic and flexible properties and are frequently used as soft segments in the synthesis of polyurethane. PLCL, in particular, offers superior compliance and more closely emulates the characteristics of natural blood vessels [20,21,22], making it a suitable scaffolding material for PLCL-polyurethane small-diameter artificial blood vessels [23,24,25]. Additionally, isocyanates, known for their strong mechanical properties, are commonly employed as hard segments. By adjusting their chemical composition and molecular structure, the mechanical properties and degradation rate of the material can be enhanced [26]. Hydrogen bonds are frequently incorporated into both the polymer backbone and side chains to enhance the mechanical properties of polymers through intra- and intermolecular noncovalent interactions [27,28,29]. Various design strategies for introducing hydrogen bonding into the side chain have been documented, including the incorporation of amide, urea, and thymine groups. However, the majority of hydrogen bonds utilized thus far are mono-(amide) and di-(urea) bonds, which generate limited intermolecular forces [30,31,32]. Given that urea-pyrimidinone (UPy) possesses tetravalent hydrogen bonding motifs, linking UPy to polyurethane (PU) and incorporating it into the PLCL backbone could serve as an effective strategy to enhance the mechanical strength and elastic modulus of PLCL-PU artificial blood vessels [33,34,35].

Small-diameter vascular grafts often experience failure in clinical applications due to thrombosis and inadequate biocompatibility [36]. Common heparin (Hep), a widely studied glycosaminoglycan, is renowned for its anticoagulant properties; however, it is associated with significant side effects, including an elevated risk of bleeding [37]. Recent studies have demonstrated that low-molecular-weight heparin (LHep) offers notable advantages over traditional heparin, such as improved predictability [38,39], dose-dependent plasma levels, an extended half-life, and reduced bleeding risk for equivalent antithrombotic efficacy. Additionally, LHep presents a lower risk of osteoporosis with prolonged use and a diminished likelihood of heparin-induced thrombocytopenia and thrombosis, which are potential adverse effects of heparin [22,37,40]. Consequently, the functionalization of LHep and biocompatible gelatin (Gel) [41,42,43] on artificial blood vessels reinforced by multiple hydrogen bonds may result in a vascular graft exhibiting superior anticoagulant properties, enhanced biocompatibility, and improved mechanical characteristics [44,45].

In this study, we developed a polyurethane-based artificial blood vessel, designated as PIIU-B@LHep/Gel, which is reinforced by multiple hydrogen bonds and dual-functionalized with low-molecular-weight heparin and gelatin to enhance both blood compatibility and biocompatibility (refer to Figure 1). The synthesis of urea-pyrimidone (Upy)-based polyurethane elastomers (PIIU-B) was achieved by incorporating quadruple hydrogen bonds within the polymer backbone, utilizing poly(L-lactide-co-ε-caprolactone) (PLCL) as the soft segment and isophorone diisocyanate (IPDI) and di-(isophorone diisocyanate)-Ureido pyrimidinone (IUI) as the hard segments. The PIIU-B was subsequently fabricated into a small-diameter artificial blood vessel through electrospinning, with optimization of the IUI/IPDI composition. The incorporation of multiple hydrogen bonds significantly enhanced the tensile strength and elastic modulus of the vessels. Following this, the PIIU-B@LHep/Gel vessel underwent dual-functionalization with low-molecular-weight heparin and gelatin via UV crosslinking. Gelatin (Gel) was employed to promote the aggregation and adhesion of vascular endothelial cells (VECs), while low-molecular-weight heparin (LHep) served to inhibit platelet (PLT) aggregation, thereby reducing the risk of thrombosis. The resulting vessel demonstrated excellent anticoagulant properties and superior biocompatibility. In vivo experiments were conducted using SD rat models to evaluate the performance of the developed vessel.

## 2. Materials and Methods

All chemical reagents were procured from Tianjin Xiens Biochemical Technology Co., Ltd. (Tianjin, China). Mouse fibroblasts (L929) were obtained from the Cell Bank of the Chinese Academy of Sciences. The L929 cells were cultured in Dulbecco’s Modified Eagle Medium (DMEM) supplemented with 10 wt% fetal bovine serum (FBS), 100 U/mL penicillin, and 100 U/mL streptomycin, maintained in a cell incubator at 37 °C with 5% CO_2_. Male SD rats were acquired from Beijing Viton Lihua Laboratory Animal Technology for subcutaneous implantation studies. All SD rats were housed in specific pathogen-free (SPF) animal facilities, adhering strictly to the International Code of Ethics and the National Institutes of Health guidelines for the care and use of laboratory animals. The animal experiments received approval from the Animal Protection Ethics Committee of the Institute of Radiation Medicine, Chinese Academy of Medical Sciences (Approval No. SYXK (Jin) 2019-0002).

### 2.1. Synthesis and Characterization of IUI

For the synthesis of IUI, 2 mL 2-acetylbutyrolactone and 3.3 g guanidine carbonate were dissolved in 20 mL anhydrous ethanol, and 5.2 mL triethylamine was added and refluxed at 70 °C overnight until the solution turned yellow and cloudy. The solid was filtered, washed, and redissolved. The pH was set to 6–7 using a dilute hydrochloric acid solution for precipitation and drying to obtain Upy, which then reacted with isophorone diisocyanate (IPDI) with a 1:12 molar ratio at 90 °C under the protection of argon gas for 3 days. n-Hexane was added to precipitate a white colloidal solid [46].

### 2.2. Synthesis of PIIU-B Polyurethane Elastomers

PIIU-B elastomers were synthesized using poly-lactide-L-polycaprolactone (PLCL, 50/50) as the soft segment, IUI and IPDI as the hard segment, and butane-1,4-diol (BDO) as a chain extender, and the product was named PIIU-B. In addition, the PIU-B elastomers were named PLCL as the soft segment, IUI as the hard segment, and BDO as the chain extender. The detailed experimental steps were as follows: 5 g of PLCL (Mn = 2000, LA/CL = 50/50) was vacuum-heated and melted at 120 °C for 20 min and then cooled to 80 °C while adding 15 mL of ultra-dry DMF under the protection of argon; then, 1.5 g of IUI and 3 drops of dibutyltin dilaurate (DBTDL) were added to PLCL solution, and the reaction was kept under the protection of argon at 80 °C for 16 h. Then, 223 μL of 1,4-butanediol (BDO) was added, and we continued to stir at 80 °C for 6 h; the product was dialyzed in distilled water (3500 Da) for 3 days and freeze-dried to obtain PIIU-B elastomers. According to different ratios of IUI and IPDI, a series of PIIU-B elastomers was prepared; those were PIIU-B (IUI:IPDI = 0.3:1), PIIU-B (IUI:IPDI = 0.5:1), PIIU-B (IUI:IPDI = 1:1), and PIIU-B (IUI:IPDI = 2:1).

The chemical structures of Upy, IUI, PIU-B, and PIIU-B were characterized by high-resolution mass spectrometry (HRMS, Thermo Scientific QE plus), infrared absorption spectroscopy (FT-IR, Bruker Vertex 70, Germany), nuclear magnetic resonance hydrogen spectroscopy (^1^H-NMR, Bruker Avance III HD 500 MHz, Germany), Gel Permeation Chromatography (GPC, Agilent GPC 50, USA), Differential scanning calorimetry (DSC, TA Q2000, USA), and a wide-angle X-ray diffraction pattern (WAXRD, Rigaku Ultima IV, Japan).

### 2.3. Preparation and Characterization of PIIU-B Artificial Blood Vessels

PIIU-B artificial blood vessels with different IUI to IPDI ratios were prepared by electrospinning [47]. The PIIU-B polyurethane was dissolved in hexafluoroisopropyl alcohol with a concentration of 20%, and the spinning conditions were set as follows: the applied voltage was 18kV, the syringe pushing speed was 1.0 mL/h, and the distance of the needle-to-collector was 15.0 cm; the fibers obtained under this condition were randomly distributed without orientation. Electrospun nanofibers were collected into tubular fabrics by means of stainless-steel rods (diameter 4 mm, length 300 mm, speed 500 r/min) or aluminum-receiving cylinders (diameter 100 mm, length 300 mm, speed 500 r/min). The morphology and structure of the PIIU-B vessels, with varying IUI/IPDI ratios, were characterized using a scanning electron microscope (SEM, TESCAN MIRA LMS, Czech Republic). The diameter and porosity of the PIIU-B vessels were quantified using ImageJ software (Java 1.8.0_345). Mechanical tensile testing was performed in an ambient air environment, and the mechanical properties of the PIIU-B vessels were evaluated using a universal tensile testing machine. The elastic modulus, maximum tensile stress, and elongation at break were calculated. Data were presented as the mean ± standard deviation (n = 3).

### 2.4. Synthesis and Characterization of LHepMA and GelMA

High-molecular heparin (5 g) was nitrite hydrolyzed by sodium nitrite (6.9 mg/mL) in water (50 mL) at different conditions (pH4.0/0 °C, pH2.5/25 °C, and pH4.0/25 °C) to obtain the low-molecular-weight heparin with aldehyde (LHep-CHO), and the reaction was stopped by adjusting the pH to 8.0 with a 1M NaOH solution. LHepCHO with different compositions was incubated with a Schiff reagent (5 wt% and 50 μL), and the color change in the Schiff solution qualitatively evaluated the aldehyde group. Then, LHep-CHO (200 g) reacted with amino methacrylide in water at a pH of 5.0 with an added 2 mg sodium cyanoborohydride (NaBH_3_CN) and 415 μL 2-aminoethyl methacrylate hydrochloride (AEMA-HCl, 1 wt%); after dialysis and freeze drying, LHepMA was obtained. The chemical structures of LHepCHO and LHepMA were verified by FT-IR, ^1^H-NMR, and GPC.

GelMA was synthesized by reacting methacrylic anhydride (MMA) with gelatin (Gel). MMA (20 mL) was added to a 10 wt% Gel solution, and the solution pH was adjusted to ~8 and allowed to react for 4 h. The product was dialyzed in water (3500 Da) for 3 days and freeze-dried to obtain methacrylated gelatin (GelMA). ^1^H-NMR was used to confirm the successful synthesis of GelMA.

### 2.5. Functionalization of LHepMA and GelMA onto PIIU-B Vessels

The PIIU-B vessels were subjected to plasma treatment at 50% rating power for 5 min after evacuation to 60 Pa to increase their hydrophilicity. Subsequently, UV crosslinking was conducted in an aqueous solution, where both the inner and outer surfaces of the PIIU-B vessels were immersed in a mixture of LHepMA (5 wt%) and GelMA (5 wt%) and exposed to UV irradiation (60 W, 5 min) for photocrosslinking, employing 2-hydroxy-4′-(2-hydroxyethoxy)-2-methylpropiophenone (I2959, 1 mg/mL) and polyethylene glycol diacrylate (PEGDA, 20 μL/mL) as a photoinitiator and chain extender, and LHepMA and GelMA functionalized as PIIU-B vessels (PIIU-B@LHep/Gel) were prepared. For thermal crosslinked PIIU-B@LHep/Gel, the crosslinked reaction was performed using ammonium persulfate (APS, 1 mg/mL) as the thermal initiator and PEGDA (20 μL/mL) as the extender at 60 °C for 30 min. Functionalization of gelatin on PIIU-B@LHep/Gel was detected using X-ray photoelectron spectroscopy (XPS, Thermo Scientific K-Alph, Shanghai, China). The heparin content of PIIU-B@LHep/Gel was determined using the toluidine blue method. The morphology and structure of PIIU-B@LHep/Gel were observed using SEM. The hydrophilicity of PIIU-B@LHep/Gel was assessed using a Contact Angle meter (JC2000FM; Shanghai Zhongchen Digital Technology Instrument Co., Ltd., Shanghai, China).

### 2.6. Anticoagulant Activity of PIIU-B@LHep/Gel Vessels

In vitro coagulation experiments were conducted to characterize the anticoagulation ability. The tested materials (PLCL, PIIU-B, PIIU-B@LHep/Gel, PLCL + 5%Hep) had a diameter of 6.5 mm and were placed in a 96-well plate. Then, 50 μL of fresh SD rat blood was added to each well. After the scheduled time points (1, 2, 3, 4, 5, 10, 15, 20, 25, and 30 min), all soluble blood components were removed and the tested samples were photographed. Platelet adhesion experiments were performed, and 10 mL of fresh SD rat blood was centrifuged at 500 G for 10 min and 800 G for 10 min, and the lower 1/3 of the supernatant was taken as platelet-rich plasma (PRP). The tested materials (PLCL, PIIU-B, PIIU-B@LHep, and PIIU-B@LHep/Gel) were incubated in 1 mL of PRP at 37 °C for 1 h; then, PRP was removed, and 2.5 wt% of glutaraldehyde was added for further incubation at 25 °C for 2 h, and the platelet adhesion was observed by SEM.

The anticoagulation effect was quantified using the activated partial thromboplastin time (APTT) test and thrombin time (TT) tests. Fresh SD rat venous blood (2 mL) was collected by vacuum blood collection with sodium citrate (Jiangsu Kangjian Medical Supplies Co., Ltd., Nanjing, China), centrifuged for 10 min at 3000× *g* rpm, and the plasma was collected. PIIU-B, PIIU-B@LHep, and PIIU-B@LHep/Gel were incubated with plasma; 0.1 mL of an APTT reagent and 0.1 mL of a CaCl_2_ reagent were added to 0.1 mL of the treated plasma and incubated at 37 °C for 5 min. APTT was recorded when fibrin filaments were observed. The material-treated plasma was taken, 0.1 mL of thrombin was incubated at 37 °C for 5 min, and the coagulation TT time was recorded. All experiments were repeated three times.

### 2.7. In Vitro Biocompatibility of PIIU-B@LHep/Gel Vessels

The blood compatibility of PIIU-B@LHep/Gel was evaluated using the hemolysis test: freshly collected blood from SD rats was collected, centrifuged at 1500 rpm for 5 min, and resuspended in PBS to obtain a 20% erythrocyte solution. PLCL, PIIU-B, and PIIU-B@LHep/Gel were incubated with an erythrocyte solution for 1 h at 37 °C to assess their blood compatibility, and the water incubated erythrocyte solution was used as the positive group; the PBS-incubated erythrocyte solution was used as the negative group. The samples were photographed, the absorbance of the erythrocyte solution was measured at 540 nm using a microplate reader, and the hemolysis rate was calculated. The calculation method of hemolysis rate was calculated according to the following formula [48]:$$\mathrm{Hemolysis}\;\mathrm{rate}\;(\%)=\frac{OD_{Experiment}-OD_{Negative}}{OD_{Positive}-OD_{Negative}}\times 100\%$$

Among them, *OD_experiment_*, *OD_negative_*, and *OD_positive_* represented the absorbance at 540 nm with a microplate reader for the experimental group, PBS group, and water group, respectively.

The cell compatibility was assessed on L929 cells by a dead and live staining kit. Mouse fibroblast (L929) cells were purchased from the cell bank of the Chinese Academy of Sciences. L929 cells were cultured in a Dulbeck Modified Eagle Medium (DMEM) supplemented with 10 wt% fetal bovine serum (FBS), 100 U/mL penicillin, and 100 U/mL streptomycin in a cell incubator (37 °C, 5% CO_2_). PLCL, PIIU-B, PIIU-B@Gel, and PIIU-B@LHep/Gel were immersed in PBS at 37 °C for 48 h. Leaching liquors of different materials were used to culture L929 cells for one and three days. Cells were stained using a live/dead kit and photographed under a fluorescence microscope. The cell compatibility of the PIIU-B@LHep/Gel was also detected using the Cell Counting Kit-8 (CCK-8), and cell viability was tested and analyzed.

### 2.8. Subcutaneous Implantation Experiments

In this study, 54 male SD rats (8 weeks of age) were purchased from Beijing Viton Lihua Laboratory Animal Technology Co for subcutaneous implantation, with a single rat as an experiment unit. The rats were divided into nine groups: PLCL, PIIU-B, and PIIU-B@LHep/Gel samples, with embedding times at 7, 14, and 28 days. Among these, PLCL and PIIU-B groups at different times were served as the compare group. To ensure that the rats of each group were ≥5 at the final test, we set the rats to 6 per group to consider possible accidental infections and death. When the animals were infected or died, except for embedding reasons, they were excluded from experimental statistics. Hence, six rats were allocated to each group, with a total of nine groups and a total of fifty-four rats (6 rats in each group × 9 groups) in this experiment. The mice were randomly divided into these groups, numbered from 1 to 54, and the experimental groups were randomly assigned mouse numbers using Excel. The rats under study were housed in SPF animal houses (one rat in one cage) and benefited from regular care and feeding conditions, which included a temperature of 23 °C, humidity of 55%, and light/dark cycles of 12 h each. To prevent experimental confusion, the rat cages were labeled with different materials and colors and placed in different positions at different times. The experimental groups of 7 days, 14 days, and 28 days were placed on the first, second, and third layers of the mouse rack, with colors of PLCL (yellow) and PIIU-B (blue), respectively, and PIIU-B@LHep/Gel (red). Jimin Zhang and Jingzhe Guo were aware of the group allocation at the different stages of the experiment. During the experiment, anesthesia was used to reduce pain, suffering, and distress in the rats, and we reported and recorded any expected or unexpected adverse events in the experiment.

To investigate the in vivo histocompatibility of PIIU-B@LHep/Gel, artificial blood vessel materials were subcutaneously implanted for verification purposes [49]. A 2 cm incision was made along the mid-dorsal axis of the rats to create subcutaneous pouches, into which PLCL, PIIU-B, and PIIU-B@LHep/Gel (with a wall thickness of 250 μm, an inner diameter of 4 mm, and a length of 2 cm) were respectively embedded. Observations were conducted on days 7, 14, and 28 post-implantations in a controlled animal operating room environment. Upon completion of the experiment on day 28, the animals were euthanized using an overdose of anesthesia followed by cervical dislocation. The implanted materials and adjacent tissues were surgically resected and collected. Subsequently, tissues corresponding to the same embedded materials were homogenized by a different experimenter for subsequent analysis. Optical and SEM images of implanted vessels were obtained. The collected tissues were fixed with 10% formaldehyde, dehydrated using an ethanol gradient, and embedded in paraffin. The paraffin-embedded tissues were cut into 5 μm thick sections and stained with hematoxylin-eosin (H&E), CD68 immunofluorescence, and Masson trichrome (MT) for evaluating the in vivo biocompatibility of PIIU-B@LHep/Gel vessels.

### 2.9. Data Analysis

There were no exclusions in the in vivo experiment. For each analysis, the exact value of n was 6. All the data presented in this paper were presented in the form of mean ± standard deviation (SDs), and the differences between the two or more groups were evaluated by *T*-test or one-way ANOVA using GraphPad software. Significance levels were marked with * *p* < 0.05, ** *p* < 0.01, *** *p* < 0.001.

## 3. Results

### 3.1. The Analysis of Synthesis and Characterization of IUI

The IUI monomer was prepared using a two-step method (Figure 1a). Guanidine carbonate was reacted with 2-acetylbutyrolactone to obtain ureido-pyrimidinone (Upy, yellow powder, yield of 86.7%), which was subsequently condensed with isophorone diisocyanate (IPDI) to yield the hydrophobic di-(isophorone diisocyanate)-ureido pyrimidinone (IUI, white powder, yield of 62.5%). Mass spectrometry revealed an ion fragment peak of IUI at 614.37, consistent with the theoretical molecular weight of 613 (Figure 1b). The FT-IR results of IUI are shown in Figure 1c, which shows an obvious strong absorption peak of -N=C=O at 2260 cm^−1^, the stretching vibration of C-N stretching, an N-H bending vibration, and a C=O stretching vibration at 1255, 1585, and 1695 cm^−1^, without an absorption peak of the hydroxyl and the amino groups, which confirmed the chemical structure of IUI. ^1^H-NMR spectra are shown in Figure 1d, compared with a characteristic peak of hydroxyl at 3.35–3.20 ppm (marked m) for Upy; that peak for IUI disappeared, and there appeared new characteristic peaks of the pyrimidine ring (marked c), secondary amine proton peak (marked b), and the secondary amine proton peak of the amide bond (marked a) of IUI at 12.95–12.85 ppm, 12.00–11.90 ppm, and 10.20–10.15 ppm, respectively, successfully verifying the synthesis of IUI.

### 3.2. The Analysis of Synthesis of PIIU-B Polyurethane Elastomers

PIIU-B elastomers with varying compositions were synthesized utilizing PLCL (molecular weight of 2000, LA/CL ratio of 50/50) as the soft segment, while IUI and IPDI served as the hard segments, and BDO functioned as the chain extender (refer to Figure 2a). The resulting elastomers were designated as PIU, PLCL-IPDI, PIIU, and PIIU-B, respectively. The chemical structures of PIU, PLCL-IPDI, PIIU, and PIIU-B were first studied by ^1^H-NMR (Figure 2b). The characteristic peaks of IUI (marked e, d, f) appeared at 12.95–12.85, 12.00–11.90, and 10.20–10.15 ppm, and the characteristic peaks of IPDI that appeared at 2.90–2.83 (marked a) and 8.02 ppm (marked c) were also found in PIIU and PIIU-B elastomers. Agreeing with the characteristic peak at 3.60–3.70 ppm attributed to the terminal hydroxyl proton of BDO (marked b) in PIIU-B, these results indicated the successful synthesis of PIIU-B elastomers. The FT-IR spectrum confirmed the chemical structure of PIIU-B, as the disappearance of the peak at 2260 cm^−1^ was attributed to the strong characteristic absorption of the -N=C=O group in PIIU-B compared with that of IUI (Appendix A). Gel Permeation Chromatography (GPC) was used to characterize the molecular weight of the synthesized polymer (Figure 2c and Appendix A), with a molecular weight of PIIU-B of 34,101 g/mol, which was significantly higher than those of PLCL (1994 g/mol), PIU (22,158 g/mol), PIIU (18,811 g/mol), and PIU-B (19,095 g/mol), further verifying the successful chain expansion and polymerization of PIIU-B.

Differential Scanning Calorimetry (DSC) analysis was employed to assess the thermal stability of polyurethane. Figure 2d presents the DSC results for PIIU-B, revealing a glass transition temperature (Tg) of 54.62 °C and a melting temperature exceeding 100 °C, which suggests the thermal stability of PIIU-B. The crystallinity of PIIU-B was evaluated using wide-angle X-ray diffraction (WAXRD) patterns, as depicted in Figure 2e. A diffraction peak at 2θ = 21.6° was observed for PLCL, IUI, PIU, and PIIU-B, with PIIU-B exhibiting the weakest peak, indicating the lowest crystallinity and suggesting enhanced processability. Consequently, PIIU-B was selected as the material for electrospinning, and the ratio of IPDI to IUI was optimized based on the spinning performance.

### 3.3. Preparation and Optimization of PIIU-B Artificial Blood Vessels

Artificial PIIU-B blood vessels were fabricated using the electrospinning technique. These vessels appeared as white structures with an internal diameter of 4 mm and a wall thickness of 250 ± 50 μm (Figure 3a,b). To optimize the pore structure and mechanical properties of the PIIU-B vessels, a series of PIIU-B polymers with IUI:IPDI ratios of 0.3:1, 0.5:1, 1:1, and 2:1, but with comparable molecular weights of 33,570, 31,731, 34,101, and 31,985 g/mol, respectively, were synthesized (Figure 3c and Appendix A) and subsequently processed into small-diameter conduits via electrospinning. The surface morphologies of these PIIU-B vessels, along with a control PLCL vessel, were examined using scanning electron microscopy (SEM). As depicted in Figure 3d, the electrospun PLCL vessels consisted of nanofibers that were neither smooth nor uniform, likely due to the rapid evaporation of the solvent hexafluoroisopropanol, which had a relatively low boiling point of 59 °C. Consequently, these nanofibers were randomly distributed without any specific orientation. The electrospun PIIU-B vessel exhibited numerous smooth nanofibers akin to the PLCL vessel. At IUI:IPDI ratios of 0.3:1 and 0.5:1, there was a widespread distribution of substantial droplets and melted fibers, with only a limited presence of nanofibers. When the IUI:IPDI ratio was increased to 1:1, the droplets vanished, and numerous nanofibers with heterogeneous diameters were observed, demonstrating a uniform fiber diameter of 0.58 ± 0.20 μm and high porosity of 41.31%. Upon further increasing the IUI:IPDI ratio to 2:1, the nanofibers became uniformly distributed across the surface, with the resulting PIIU-B vessels exhibiting the largest fiber diameter of 1.56 ± 0.22 μm and the highest porosity of 57.46% (Figure 3e,f). The morphology of ECM-mimetic nanofibers produced by electrospinning, including the alignment, size, composition, and modulus, can be readily adjusted to emulate native ECM by modifying the parameters of the electrospinning process. The morphology of electrospun nanofibers is capable of influencing cellular migration [50]. For instance, when cells were cultured on randomly deposited fibers, they exhibited a stellate-like morphology and became locally immobilized by the fibers. This interaction led to the significant spatial organization and remodeling of the structures in contact with the cells, facilitating tissue regeneration. In stark contrast, when cells were cultured on aligned nanofibers, they rapidly oriented themselves along the fibers. Time-lapse recordings demonstrated pronounced cellular movements confined to the direction of the fibers, resulting in the production of a highly aligned collagen matrix. This alignment is advantageous for the repair of tissues with anisotropic anatomies, such as tendons, cardiac tissue, and nerve tissues [51,52]. Consequently, the randomly oriented ECM-mimetic nanofiber morphology in PIIU-B could effectively regulate the spatial adhesion and infiltration of endothelial and smooth muscle cells, promoting vascular regeneration. The nanofibers’ structure and high porosity are conducive to cell adhesion and infiltration, which are beneficial for the tissue engineering of blood vessels. Furthermore, an analysis was conducted on the varying pore sizes of artificial blood vessels, specifically examining PIIU-B electrospinning tubes with IUI:IPDI ratios of 0.3:1, 0.5:1, 1:1, and 2:1, which exhibited pore sizes of 6.59 μm, 4.92 μm, 3.89 μm, and 6.92 μm, respectively (refer to Appendix A). According to previous studies, the three-dimensional geometrical parameters, such as pore size and porosity, of artificial blood vessels significantly influence cell adhesion. A smaller pore size may restrict cell extension, resulting in reduced cell adhesion, whereas a larger pore size with curved configurations may facilitate the formation of more extensive cell adhesions [53,54,55]. Research has demonstrated that a pore size of approximately 5 μm and a porosity of around 40% markedly enhance cell infiltration in electrospun artificial blood vessels [56,57]. Consequently, the PIIU-B electrospinning tubes possess optimal pore size and porosity characteristics conducive to cell infiltration and tissue regeneration in artificial blood vessels.

To facilitate the regeneration of vascular tissue, electrospun vessels must offer mechanical support while also satisfying requirements for suture retention and blood pressure tolerance. Consequently, there is an urgent need for vessels with superior mechanical properties. As tensile stress-strain curves shown in Figure 3g, compared to the elastic modulus of PLCL (1.30 MPa), those for PIIU-B were significantly improved, and the values for IUI:IPDI of 0.3:1, 0.5:1, 1:1, and 2:1 were 1.71, 3.64, 12.45, and 7.98 MPa, respectively (Figure 3h). As the IUI: IPDI ratio increased from 0.3:1 to 1:1, the elastic modulus of PIIU-B increased owing to the strengthening effect of the quadruple hydrogen bond interactions between the IUI molecules. When IUI:IPDI was up to 2:1, a decrease in the elastic modulus was observed. The reason for PIIU-B with IUI:IPDI 1:1 has the highest elastic modulus might be explained as follows: as the proportion of IUI increases in PIIU-B, the hydrogen bond density increases, resulting in an increase in elastic modulus; when the ratio of IUI to IPDI is increased to 1:1, the elastic modulus was up to maximum value; when we further increased the proportion of IUI to 2:1, the greater the strengthening effect of quadruple hydrogen bonds, which makes the material become brittle. Moreover, the maximum tensile stress for IUI:IPDI of 0.3:1, 0.5:1, 1:1, and 2:1 were 8.71, 7.48, 5.28, and 3.41 MPa, respectively (Figure 3i).

The maximum tensile strength at fracture (maximum load) for artificial blood vessel materials with IUI:IPDI ratios 0.3:1, 0.5:1, 1:1, and 2:1 were tested; those values were 9.97 N, 11.30 N, 22.78 N, and 17.36 N, respectively. Among these, artificial blood vessel materials with a IUI: IPDI ratio of 1:1 showed the highest maximum load, indicating the strongest mechanical strength (Appendix A). The elongations for IUI:IPDI of 0.3:1, 0.5:1, 1:1, and 2:1 were 537.6%, 361.3%, 150.3%, and 101.2%, respectively (Appendix A). For the mechanical performance of PIIU-B, the elastic modulus (12.45 MPa) of PIIU-B was significantly higher than those of the reported polycaprolactone (PCL, 4.1 MPa), poly(L-lactide-co-ε-caprolactone) (PLCL, 4.69 MPa), polyurethane (PU, 8.17 MPa), and polyurethane ester urea (PEUU, 1.4 MPa). In addition, the maximum tensile stress (5.28 MPa) of PIIU-B was higher than that of PLCL (3.23 MPa) and polyurethane material PU (4.53 MPa) and comparable to that of PEUU (5.85 MPa) [24,58,59]. Therefore, the mechanical performance of PIIU-B was adequate to meet the requirements for tissue-engineered blood vessels. In consideration of micromorphology and mechanical properties, PIIU-B with an IUI:IPDI ratio of 1:1, characterized by sufficient porosity, a uniform fiber diameter, and a strong elastic modulus, was selected as the electrospun material for artificial blood vessel tissue engineering. Additionally, the oxidative degradation of PIIU-B polyurethane, with an IPDI to IUI ratio of 1:1, was assessed after one year of storage at room temperature using Gel Permeation Chromatography (GPC) to evaluate changes in its molecular weight. As illustrated in Appendix A, the molecular weight of PIIU-B was initially 34,101 g/mol, and after one year of storage at room temperature, it decreased to 29,223 g/mol. This minor degradation of PIIU-B suggests that it remains suitable for the tissue regeneration requirements of artificial blood vessels.

### 3.4. The Analysis of Synthesis and Characterization of LHepMA and GelMA

In comparison to high-molecular-weight heparin (HHep), low-molecular-weight heparin (LHep) offers several advantages, including an extended metabolic half-life, enhanced anticoagulant efficacy, and the potential for functionalization in medical devices. To achieve batch production and ensure the stability of LHep, the degradation of HHep was conducted using sodium nitrite under varying pH levels and temperatures (Figure 4a), resulting in the formation of the nitrite derivative LHepCHO. The characterization of LHepCHO was performed using ^1^H-NMR and Gel Permeation Chromatography (GPC). In the ^1^H-NMR spectra (Figure 4b), a new distinct peak corresponding to the –CH– group (marked as ‘a’) emerged at 1.3 ppm for LHepCHO prepared at pH 4.0 and 25 °C, as well as at pH 4.0 and 0 °C, when compared to HepNa and LHepCHO at pH 2.5 and 25 °C. This observation indicates the formation of the –CHO group in heparin. In GPC data (Figure 4c and Appendix A), LHepCHO-pH 4.0 at 25 °C showed the lowest molecular weight (4070 g/mol), significantly lower than that of LHepCHO-pH 4.0, 0 °C (8898 g/mol), LHepCHO-pH 2.5, 25 °C (14,021 g/mol), and HepNa (21,071 g/mol). Thus, with the reaction condition of a pH of 4.0 at 25 °C, LHepCHO with low molecular weight and abundant aldehyde could be efficiently attained after the sodium nitrite treatment of HHep, and the LHepCHO-pH 4.0 at 25 °C was designated LHepCHO in what follows.

Next, LHepMA was synthesized by the reaction between LHepCHO and N-(3-aminopropyl) methacrylamide via the linkage of the aldehyde and amino groups. This reaction could be intuitively monitored, as the aldehyde group of LHepCHO was colored by a chromogenic agent (magenta) and exhibited a pink color when the reaction changed from pink to colorless, indicating that LHepCHO was converted to LHepMA (Appendix A). The FT-IR spectra showed that LHepMA exhibited a new peak of the C=O-stretching vibration at 1758 cm^−1^ in comparison with LHepCHO (Figure 4d), suggesting the successful grafting of N-(3-aminopropyl) methacrylamide onto LHepCHO. Furthermore, ^1^H-NMR spectra confirmed the successful synthesis of LHepMA by the disappearance of the peak at 1.35 ppm (marked a) and the appearance of a new peak at 5.91 ppm (marked c) and a peak at 1.85 ppm (marked b), compared with LHepCHO (Figure 4e). Subsequently, the hydrophilic gelatin (Gel) reacted with N-(3-aminopropyl) methacrylamide to produce GelMA, which could be further modified on the surface of the PIIU-B vessels to improve biocompatibility (Figure 4f). In ^1^H-NMR spectra, we verified the successful synthesis of GelMA by the peaks at 5.91 and 6.42 ppm (Figure 4g), which were attributed to the chemical shifts of CH_2_ (marked m and n) and CH_3_ (marked p).

### 3.5. The Analysis of Functionalization of LHepMA and GelMA onto PIIU-B Vessels

LHepMA and GelMA were functionalized onto PIIU-B vessels through UV-induced crosslinking. Initially, the PIIU-B vessels underwent plasma treatment to enhance their hydrophilicity. Subsequently, LHepMA and GelMA were polymerized and crosslinked in an aqueous solution under ultraviolet (UV) light, utilizing L2959 as the photoinitiator and polyethylene glycol diacrylate (PEGDA) as the crosslinking agent (Figure 5a).

The fiber diameter and porosity before and after the UV crosslinking of PIIU-B vessels were analyzed. The results showed that there were no apparent changes before and after UV crosslinking for the PIIU-B vessel on the apparent morphology, fiber diameter, wall thickness, lumen diameter, and porosity (Appendix A). Gelatin immobilization on the PIIU-B vessel was assessed using XPS. The N1s core spectra of the PIIU-B showed two peaks of –N= (44.44%) and –NH-(55.56%), while, after gelatin functionalization, that for PIIU-B@Gel was only –NH-(100%); an increase in the –NH- ratio and a disappeared –N= bond indicated successful gelatin modification on the surface of PIIU-B@Gel (Figure 5b). Immobilized low-molecular-weight heparin was quantified using a toluidine blue colorimetric assay. As shown in Figure 5c, calculated according to the standard toluidine blue curve, the heparin density on the vessels of PIIU-B@LHep and PIIU-B@LHep/Gel were 96.5 mg/cm^2^ and 94.7 mg/cm^2^, which were much higher than those of the control (0.43 mg/cm^2^) and PIIU-B (0.52 mg/cm^2^). PIIU-B@LHep and PIIU-B@LHep/Gel could be approximated as having no heparin attached to the surface. These findings confirm the successful functionalization of LHepMA and GelMA onto PIIU-B vessels. As illustrated in the transverse and longitudinal sections in Figure 5d,e, the morphology of PIIU-B@LHep/Gel preserved the nanofiber structures of PIIU-B at the intersections, in contrast to the PIIU-B vessels with nanofiber configurations. Contact angle measurements were conducted to assess alterations in hydrophobicity and hydrophilicity following the modification of the PIIU-B surface with heparin and gelatin [60]. As depicted in Appendix A, the contact angle of PIIU-B@LHep/Gel progressively decreased over time, reaching 45.2° after a droplet residence time of 10 s, and further reducing to 3° when the droplet residence time was extended to 2 min. The water contact angle of PIIU-B@LHep/Gel was reduced from 130.8° to 3° following the LHep/Gel modification of the PIIU-B surface (Figure 5f), demonstrating a substantial enhancement in hydrophilicity [61]. Furthermore, the application of plasma processing combined with UV-crosslinking technology proved to be more effective than the absence of plasma processing or the use of thermal crosslinking technology in the preparation of PIIU-B@LHep/Gel, as evidenced by the more pronounced decrease in the water contact angle with UV-P treatment. The crosslinking of LHepMA and GelMA on the PIIU-B@LHep/Gel tube significantly enhanced hydrophilicity, which is advantageous for cell adhesion and infiltration on the scaffold, thereby contributing to the superior tissue regeneration capabilities of the artificial blood vessel.

### 3.6. The Analysis of Anticoagulant Activity of PIIU-B@LHep/Gel Vessels

To evaluate the anticoagulant properties of low-molecular-weight heparin on the PIIU-B@LHep/Gel surface, an in vitro coagulation assay was conducted using fresh rat blood to assess the anticoagulation efficacy of PIIU-B@LHep/Gel (refer to Figure 5g). In the cases of PLCL and PIIU-B, blood clotting was initiated within 2 min, with complete coagulation occurring after approximately 15 min. Conversely, the PIIU-B@LHep/Gel group exhibited no significant clot formation even after extending the observation period to 30 min, demonstrating superior anticoagulation performance compared to the PLCL + 5% heparin group. Additionally, platelet adhesion experiments were conducted and analyzed using SEM (see Figure 5h). For PLCL and PIIU-B, a significant accumulation and adhesion of platelets were observed on the surface, attributable to the microstructure of the electrospun fiber biomimetic extracellular matrix (ECM), which facilitated platelet adhesion. Conversely, in the PIIU-B@LHep and PIIU-B@LHep/Gel groups, which incorporated heparin modification, platelet adhesion was notably absent, thereby confirming the effective anticoagulation properties of PIIU-B@LHep/Gel (Appendix A). The anticoagulation efficacy was further quantified using activated partial thromboplastin time (APTT) and thrombin time (TT) assays. The data indicated that PIIU-B@LHep/Gel demonstrated the highest APTT (157.2 s) and TT (64.2 s), underscoring its superior anticoagulation capability (Figure 5i). These findings align with previous studies suggesting that heparin can inhibit coagulation by binding to antithrombin III, thereby preventing platelet aggregation and adhesion.

### 3.7. The Analysis of PIIU-B@LHep/Gel Vessels In Vitro Biocompatibility

The blood compatibility of PIIU-B@LHep/Gel was assessed through a hemolysis test (Figure 6a), utilizing water and PBS groups as positive and negative controls, respectively. In comparison to the positive control group, which exhibited a 100% hemolysis rate, PIIU-B@LHep/Gel demonstrated a significantly lower hemolysis rate of 0.03%. This rate is comparable to those observed for PIIU-B (0.23%), PLCL (0.15%), and the negative control group (Figure 6b). All hemolysis rates were below the 5% threshold stipulated by standards for vascular materials, thereby affirming the superior blood compatibility of PIIU-B@LHep/Gel as a vascular material. Given that PIIU-B@LHep/Gel artificial blood vessels are designed for long-term implantation in blood-contacting applications, excellent blood compatibility is a crucial criterion for their clinical use.

Cell compatibility was evaluated in L929 cells using a live/dead staining kit, where dead cells were labeled red and living cells green. The PIIU-B@LHep/Gel scaffold was immersed in a cell culture medium for 48 h, after which the scaffold leaching solution was collected and used to culture L929 cells for one and three days. The cells were subsequently stained with the live/dead staining kit (Figure 6c). The visual field predominantly displayed green cells with negligible red cells, and the green fluorescence intensity was similar across PIIU-B@LHep/Gel, PIIU-B@Gel, and PIIU-B, indicating favorable cell compatibility in vitro (Figure 6d). The cytotoxicity of PIIU-B@LHep/Gel was further assessed using the CCK-8 assay. To investigate cell compatibility following LHep and Gel dual-modification on vessels, we compared unmodified vessels (PIIU) and vessels modified with only Gel (PIIU-B@LHep/Gel) as the control group (Figure 6e). The PIIU-B@LHep/Gel showed the highest cell viability on days 1 (98.23%) and 3 (99.52%), and the cell viability gradually increased until day three, confirming the good compatibility of PIIU-B@LHep/Gel.

### 3.8. In Vivo Histocompatibility of PIIU-B@LHep/Gel

To assess in vivo histocompatibility, PLCL, PIIU-B, and PIIU-B@LHep/Gel vessels were subcutaneously implanted into the dorsal region of Sprague-Dawley rats. Subsequent histopathological analyses were conducted on samples collected at intervals of 7, 14, and 28 days post-implantation (Figure 7a) [62,63]. On day 7, minimal tissue adhesion was observed on the surface of the PLCL and PIIU-B electrospun tubes, while the PIIU-B@LHep/Gel electrospun tube exhibited complete coverage with thin tissue and capillary formation. By day 14, there was an increase in tissue integration within the implanted tubes, while the PLCL sample still displayed a discernible exposed tube. By day 28, the PIIU-B@LHep/Gel electrospun tubes were entirely enveloped by a thick layer of tissue, in contrast to the PLCL and PIIU-B electrospun tubes, which were only covered by a thin tissue layer (Figure 7b). The surface of the PIIU-B@LHep/Gel electrospun material was observed to be smooth, with no evidence of bacterial adhesion or colonization, as depicted in Figure 7c. This observation suggests that PIIU-B@LHep/Gel exhibits significant antibacterial properties. The antibacterial efficacy can be attributed to two main factors: firstly, the incorporation of the hydrophilic polymer gelatin on the material’s surface substantially increased its hydrophilicity [64]; secondly, the introduction of negatively charged heparin modified the surface charge characteristics of the material [65]. This UV crosslinked, dual-functionalized bioactive molecule technology offers a novel approach for the development of other antibacterial bioactive materials, such as antibacterial orthopedic implants [66]. The external surfaces of the vessel samples collected on day 28 were subjected to further analysis using SEM. As depicted in Figure 7c, the PLCL samples exhibited nearly exposed nanofibers with no tissue adhesion observed on the scaffold. In contrast, the PIIU-B samples demonstrated a minor presence of tissue coating on the nanofibers. Notably, the PIIU-B@LHep/Gel samples were entirely enveloped by tissue, with the tissue fully infiltrating the pores of the nanofibers. These findings confirm that the PIIU-B@LHep/Gel vessels possess excellent histocompatibility in vivo.

Cell infiltration was assessed using H&E staining, as illustrated in Figure 8a. On day 7, the PLCL sample exhibited a modest degree of cell infiltration within the vessels, whereas the PIIU-B and PIIU-B@LHep/Gel samples demonstrated negligible cell infiltration. Although PLCL served as a comparison for PIIU-B and PIIU-B@LHep/Gel, it possesses greater hydrophobicity. However, its fiber diameter (0.42 μm) was smaller than that of PIIU-B (0.58 μm) and PIIU-B@LHep/Gel (0.62 μm), and its maximum tensile stress (7.02 MPa) was higher than that of PIIU-B (5.28 MPa) and PIIU-B@LHep/Gel (5.34 MPa). Previous research has indicated that smaller fiber diameters and higher mechanical strength facilitate enhanced cell adhesion, as smaller fibers and increased mechanical strength provide a larger specific surface area and mechanical support for cell adhesion, thereby promoting cell proliferation and infiltration [67,68]. Therefore, the PLCL material, characterized by a smaller fiber diameter and higher tensile stress, may have facilitated more rapid cell infiltration compared to PIIU-B and PIIU-B@LHep/Gel at day 7. By day 14, the PIIU-B and PIIU-B@LHep/Gel samples exhibited an increased number of infiltrating cells within the vessels, whereas the PLCL sample did not demonstrate significant changes. By day 28, cells were extensively distributed in the PIIU-B@LHep/Gel sample, while only a limited number of cells were observed in the PIIU-B and PLCL samples. These findings corroborate the superior histocompatibility of PIIU-B@LHep/Gel vessels. Additionally, the wall thickness of the embedded transplant blood vessels was quantified using ImageJ software, as depicted in Figure 8a. As illustrated in Appendix A, the wall thickness measurements for PLCL at 7, 14, and 28 days were 204.2 μm, 172.4 μm, and 166.0 μm, respectively; for PIIU-B, they were 252.9 μm, 207.4 μm, and 204.2 μm; and for PIIU-B@LHep/Gel, they were 161.6 μm, 123.6 μm, and 96.9 μm.

The typical inflammatory cells associated with M1 macrophages in embedded tissue at days 7, 14, and 28 were identified using CD68 immunofluorescence, which emits red fluorescence. Optical images are presented in Figure 8b, where the PIIU-B@LHep/Gel groups exhibited the most intense red fluorescence within the embedded vessels on days 7, 14, and 28. After 28 days of embedding, the red fluorescence intensities for PIIU-B@LHep/Gel, PIIU-B, and PLCL were measured at 6.421%, 0.964%, and 0.124%, respectively (Appendix A). These findings suggest a significant infiltration of inflammatory M1 macrophages in the artificial blood vessels. Previous research has demonstrated that infiltration occurs in the early stages of implant integration, which is advantageous for the degradation of stents and subsequent tissue regeneration and repair. In our study, we compared various polymeric blood vessels following subcutaneous implantation, referencing relevant literature for comparison. Specifically, we evaluated the Gelatin-polytrimethylene carbonate electrospinning tube and the poly(L-lactic acid)-polyethylene glycol dimethacrylate electrospinning tube [69,70]. Our findings indicated a higher presence of macrophages, identified by the CD68 marker for the M1 or pro-inflammatory phenotype, in our PIIU-B@LHep/Gel artificial blood vessel. Macrophage infiltration is an immediate and essential response to biomaterial implantation [71], facilitating phagocytosis and the secretion of angiogenic factors that promote the healing process [72]. These results further confirm the potential of the PIIU-B@LHep/Gel to integrate with host tissue, which could be advantageous for future use as a vascular graft.

Furthermore, to evaluate the initial in vivo regeneration of the vasculature, collagen deposition surrounding the vessels was examined using Masson’s trichrome staining (Figure 8c). By day 28, a substantial number of dense and well-organized parallel bundles of collagen fibers (stained blue) were observed encircling the PIIU-B@LHep/Gel vessels. In contrast, only sparse collagen with random orientation was present around the PIIU-B and PLCL vessels. These findings suggest that PIIU-B@LHep/Gel significantly enhances vascular tissue regeneration, highlighting its considerable potential as a tissue-engineered vascular material.

## 4. Conclusions

In conclusion, we successfully developed a polyurethane-based artificial blood vessel, strengthened by urea-pyrimidone (Upy) multiple hydrogen bonds and dual-functionalized with heparin and gelatin to enhance both blood compatibility and biocompatibility. The urea-pyrimidone (Upy)-based polyurethane elastomers (PIIU-B), characterized by quadruple hydrogen bonds, were synthesized using poly(L-lactide-co-ε-caprolactone) (PLCL) as the soft segment and isophorone diisocyanate (IUI) and isophorone diisocyanate (IPDI) as the hard segments. The PIIU-B was fabricated into a small-diameter artificial blood vessel with a diameter of 4 mm, optimizing the IUI/IPDI composition to a 1:1 ratio, thereby enhancing the hydrogen bond strength. These vessels exhibited notable tensile stress and elastic modulus (12.62 MPa), an extracellular matrix-mimetic nanofiber morphology, and high porosity (41.31%). Subsequently, the PIIU-B vessel underwent dual functionalization with low-molecular-weight heparin and gelatin through UV crosslinking on the surface, which conferred excellent cell compatibility, enhanced anti-platelet adhesion properties, and improved coagulation parameters, including an activated partial thromboplastin time (APTT, 157.2 s) and thrombin time (TT, 64.2 s) in vitro. Following a seven-day subcutaneous implantation, the PIIU-B@LHep/Gel demonstrated remarkable biocompatibility, evidenced by complete tissue coverage, substantial cellular infiltration, and collagen formation in vivo. However, due to the inherent differences in the microenvironment between rats and humans, the findings of this study possess certain limitations. Nonetheless, this research offers a valuable reference for future investigations into the application of this artificial blood vessel in human subjects.

## Data Availability

Data will be made available on request.

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
