# Peer review of "Heparin and Gelatin Co-Functionalized Polyurethane Artificial Blood Vessel for Improving Anticoagulation and Biocompatibility"

_bioengineering, 2025, doi:10.3390/bioengineering12030304_

Round 1

Reviewer 1 Report

Comments and Suggestions for Authors

Introduction can be improved.

What about antibacterial/infection test. SHould be at least discussed.

Can nanoparticles improve biological properties?

Is there any option to use this material as coating for orthopedic/dental implants (see Bacterial adhesion on orthopedic implants. Advances in colloid and interface science. 2020, 283,  1-12, 

What about material charge properties?

Are there any test for material stability. Long life?

Reviewer 2 Report

Comments and Suggestions for Authors

The manuscript focused on three stages of biomaterial (vascular graft) evaluation: first, synthesis and characterization of polymers used in biomaterial; second, processing and characterization of the biomaterial; and third, biocompatibility assessment in terms of in vitro and in vivo evaluation. The research plan carried out in the study was constructed, and the results associated with the research plan were presented in this manuscript. However, some specific points to be addressed by the authors during the revision process are as follows:

-  The vascular grafts are designed to allow flow inside the tube. So, the surface to avoid thrombosis is the inner surface. But the anti-thrombogenic (heparinized) surface is the outer one. This design causes confusion.

- The language should be thoroughly checked, and necessary corrections should be made.

- The abstract did not mention the processing method (electrospinning) used to produce the vascular graft.

-  No citations were made for the methods and equations described in the manuscript.

-  Could the authors elaborate on the selection of in vivo subcutaneous implantation of the vascular graft instead of implanting at a site where the graft would directly be in contact with blood flow?

- In Figure 2A, “a” is used at two positions; the one on the PLCL portion seems to be mistakenly placed.

- The descriptions for Figure 2D should be revised since one melting peak (IUI) is evident. Also, if PLCL exhibits some crystallinity according to Figure 2E, a Tg value can be mentioned and a melting temperature over 100°C.

-  PLCL did not form a uniform and smooth fiber according to Figure 3D, which needs to be clarified (many material properties may be affected by the fiber morphology).

- Has any analysis been conducted to determine the fiber alignment?

-  After PIIU-B@LHep/Gel modification, the surface became super hydrophilic (3°). Superhydrophilic surfaces are not beneficial for cell adhesion. A balanced hydrophilicity is desired for cell adhesion. Is there a time-dependent comparative contact angle measurement available?

-  In Figure 6e, PLCL was chosen as the negative control (100%); PLCL cannot be taken as a negative control since it is a comparison group. Also, the graph is inconsistent with the result interpreted in the text. And what is the positive control?

Comments on the Quality of English Language

The language should be thoroughly checked.

Round 2

Reviewer 1 Report

Comments and Suggestions for Authors

accept

Author Response

Thank you  very much for your comments.

Reviewer 2 Report

Comments and Suggestions for Authors

The manuscript focused on three stages of biomaterial (vascular graft) evaluation: first, synthesis and characterization of polymers used in biomaterial; second, processing and characterization of the biomaterial; and third, biocompatibility assessment in terms of in vitro and in vivo evaluation. The authors revised the manuscript according to the reviewers’ comments. Also, they provided answers to the previous issues raised in the review report. This revision has significantly improved the article, and the authors responded properly to the comments. However, the issue in cytotoxicity assessment is still problematic since the experimental design was not properly constructed initially. According to the cytotoxicity test results presented, PIUU-B@LHep/Gel can be concluded to exhibit a tumorigenic effect (150%) when PLCL is the control group. Therefore, the authors shall find a way to present the results appropriately. The supplementary file does not contain the revised image S11.D.
